# Venus' light slab hinders its development of planetary-scale subduction

Junxing Chen [1] ✉, Hehe Jiang[1,2], Ming Tang[3], Jihua Hao[4], Meng Tian[5] & Xu Chu [1]

Terrestrial planet Venus has a similar size, mass, and bulk composition to Earth. Previous studies proposed that local plume-induced subduction existed on both early Earth and Venus, and this prototype subduction might initiate plate tectonics on Earth but not on Venus. In this study, we simulate the buoyancy of submerged slabs in a hypothesized 2-D thermo-metamorphic model. We analyze the thermal state of the slab, which is then used for calculating density in response to thermal and phase changes. The buoyancy of slab mantle lithosphere is primarily controlled by the temperatures and the buoyancy of slab crust is dominated by metamorphic phase changes. Difference in the eclogitization process contributes most to the slab buoyancy difference between Earth and Venus, which makes the subducted Venus' slab consistently less dense than Earth's. The greater chemical buoyancy on Venus, acting as a resistance to subduction, may have impeded the transition into self-sustained subduction and led to a different tectonic regime on Venus. This hypothesis may be further tested as more petrological data of Venus become available, which will further help to assess the impact of petro-tectonics on the planet's habitability.

Venus is regarded as an Earth-twin, having a slightly smaller radius (6052 km) and lower gravity ($g$ = 8.87 m/s$^2$)[1]. Despite these similarities, Venus features a $CO_2$-rich (>90%) atmosphere ($4.8 \times 10^{20}$ kg)[2] -100 times denser than Earth's ($5.15 \times 10^{18}$ kg)[3] and runaway greenhouse climate, although some researchers speculate that early Venus might have had a liquid ocean and mild surface temperature[4]. The planets' carbon budgets are primarily governed by the long-term interactions between the atmosphere and lithosphere, which are strongly associated with the tectonic regime of a planet.

Before primitive plate tectonics emerged, both planets shared stagnant/sluggish lid tectonics with limited plate mobility and material recycling[5–8]. On Venus, the distinctive corona structures are interpreted to reflect ongoing plume upwelling[7–9], and their topographic and gravimetric characteristics disclose possible local subduction[10].

Plume upwelling was also prevalent on Archean Earth, but how this regime transitioned to plate tectonics remains enigmatic[11,12]. Among the proposed models, some geodynamic studies suggest plume-initiated proto-subduction[13–15]—the rising plume broke the lithosphere and created damage zones, where the recovery was slow in a cool and water-present environment[16–18] (Fig. 1a). The loading of emplaced magma bent the lithosphere front that later formed a subducting slab (Fig. 1b), and the accumulation of such process is suggested to give birth to larger-scale subduction[19,20], dating back to as early as the Neoarchean on Earth[21]. By contrast, Venus' tectonic evolution was diverted from that of Earth, with planetary-scale subductions and plate tectonics being absent[22].

There comes the question as to why the early Venus' local plume-induced initiations failed to develop into larger-scale subductions like

[1]Department of Earth Science, University of Toronto, Toronto, Ontario M5S 3B1, Canada. [2]State Key Laboratory of Lithospheric Evolution, Institute of Geology and Geophysics, Chinese Academy of Sciences, Beijing 100029, China. [3]Key Laboratory of Orogenic Belt and Crustal Evolution, MOE; School of Earth and Space Science, Peking University, Beijing 100871, China. [4]Deep Space Exploration Laboratory/CAS Key Laboratory of Crust-Mantle Materials and Environments, School of Earth and Space Sciences, University of Science and Technology of China, Hefei 230026, China. [5]Center for Space and Habitability, Universität Bern, Bern 3012, Switzerland. ✉e-mail: junxing.chen@mail.utoronto.ca

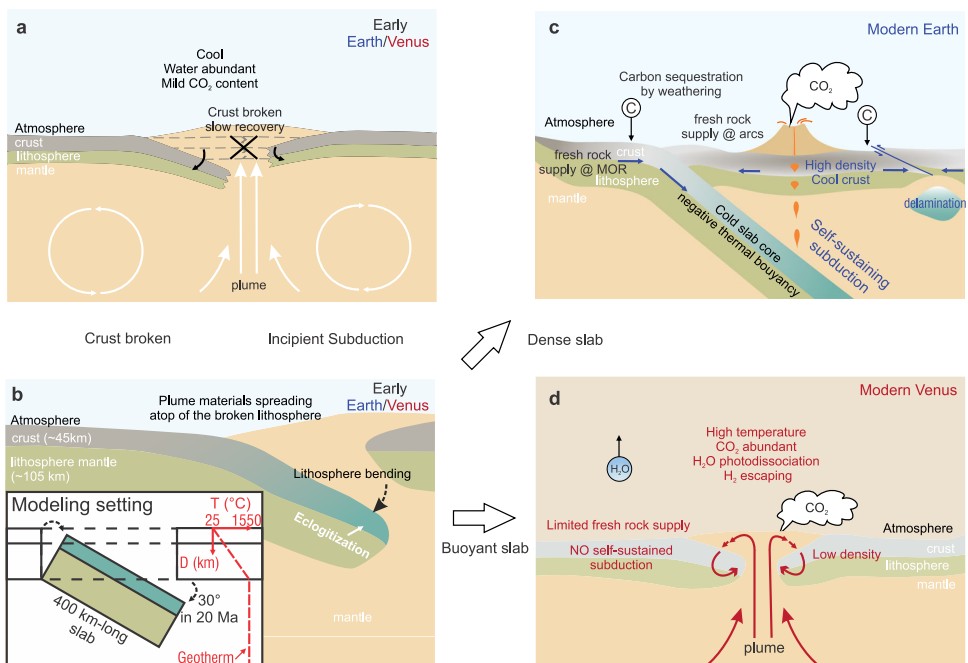

**Fig. 1 | Schematic cartoons illustrating tectonic evolutions on Earth and Venus, and their diverging carbon cycles. a** Plume upwelling breaks the lithosphere. **b** Incipient slab dipping on Venus and Earth; eclogitization might trigger self-sustaining subduction. The inset diagram displays the subduction setting and geotherm in our modeling. **c**, **d** The conceptual difference in Earth's and Venus's tectonic evolution as a result of subduction self-sustainability in different environments (see text for discussions). MOR mid-ocean ridge.

the ones on Earth. Given similar thermal states and convection dynamics in the mantles (see below), the buoyancy of the two planets' plume-induced subducting slabs provides an intuitive perspective into this question—a denser slab is less resistant to sinking, and thus its subduction more likely becomes self-sustained[20]. When a slab is submerged into the mantle, the cold slab core and dense metamorphic slab (eclogites) both contribute to the negative buoyancy for continued slab sinking and consequent plate convergence.

In this study, we investigate the density difference of Earth and Venus' subducting slabs through 2D thermal and phase-equilibria models that are similar to ref. 23. We first model a simplified pressure–temperature (P–T) field in the sinking slabs, which provides a basis for estimating the densities of the slab lithospheric mantle and crust. We then compare the negative thermal buoyancy dominated by the lithospheric mantle with the chemical buoyancy from the metamorphosed mafic slab crust. For the crust, the high-pressure metamorphic assemblages and the corresponding crust densities are determined by the evolving P–T conditions in a forward phase-equilibria simulation. We argue that the buoyancy difference between Earth's and Venus' slabs is mainly contributed by the eclogitized crust in response to different crustal compositions, and might contribute to the divergence of Earth and Venus' tectonic and environmental evolutions.

## Results and discussion
### Model setting and simplifications
We simplified the plume-induced subduction process to a slab rotating at a fixed hinge, as the lateral slab motion is minor without plate tectonics (Fig. 1b inset). The similarities of the moments of inertia and the average densities adjusted to the sizes between Venus and Earth allow us to presume that the mantle density of Venus approaches the Preliminary Reference Earth Model (PREM); the internal structure, together with the similar basaltic crusts as boundary layers, suggests similar thermal structures and convective dynamics of the mantles[24]. Thus, we assumed similar tectonic dynamics in their early ages and used the

same physical parameters, including slab length, crust and slab thickness, sinking rate and subduction depth in the models. We note that these parameters have large uncertainties due to few direct constraints. The influence of their variations on the interpretation is discussed below.

The knowledge of slab structure is rather limited on early Earth, let alone Venus. On Earth, based on postulated mantle potential temperature and associated rheological properties, the thermo-chemically stable lithosphere can be up to ~160 km thick in the Archean. On the other hand, a necessary condition for subduction initiation requires that the lithosphere is thicker than ~130 km[25] so that the net buoyancy of the lithosphere could be negative. The crustal thickness varies between 40 and 55 km according to the geochemical proxy of Archean zircons[26], in agreement with the rock strength of Archean crust[27]. On Venus, topographic and gravimetric data, as well as geodynamic models, imply a 75–150 km thick lithosphere[7] with a crustal thickness of up to 45 km in coronae[28,29]. A thin elastic layer in the lithosphere and accordingly a low crust/mantle ratio favor plume penetration and ephemeral subduction at hotspot locations[30]. The sizes of corona structures vary between 60 and >1000 km[9,31,32]. In this study, we modeled 150 km-thick, 400 km long slabs with the top 45 km being mafic crust.

In response to magmatic loading, the sinking of a slab into a viscous mantle can be regarded as a reverse process of post-glacial rebound that operates at a rate of several to dozens of millimeters per year[33,34]. We, therefore, set 10 mm/yr as the slab sinking rate in our modeling. According to modeling and gravity anomalies across coronae, the slap dip reaches 30–90°[7]. When our 400-km long slab rotates to 30°, the frontal half reaches 100–200 km, where metamorphism suffices self-driven subduction according to petrological studies on high-pressure metamorphic terranes[35,36]. We modeled the slab rotation between 0° and 30° at a sinking rate of 10 mm/yr to examine the slab buoyancy if it is forced to submerge; the subduction angle thus reaches 30° within 20 Ma.

We modeled the temperature field of the slab using a simplified heat conduction model and the Crank-Nicolson finite difference

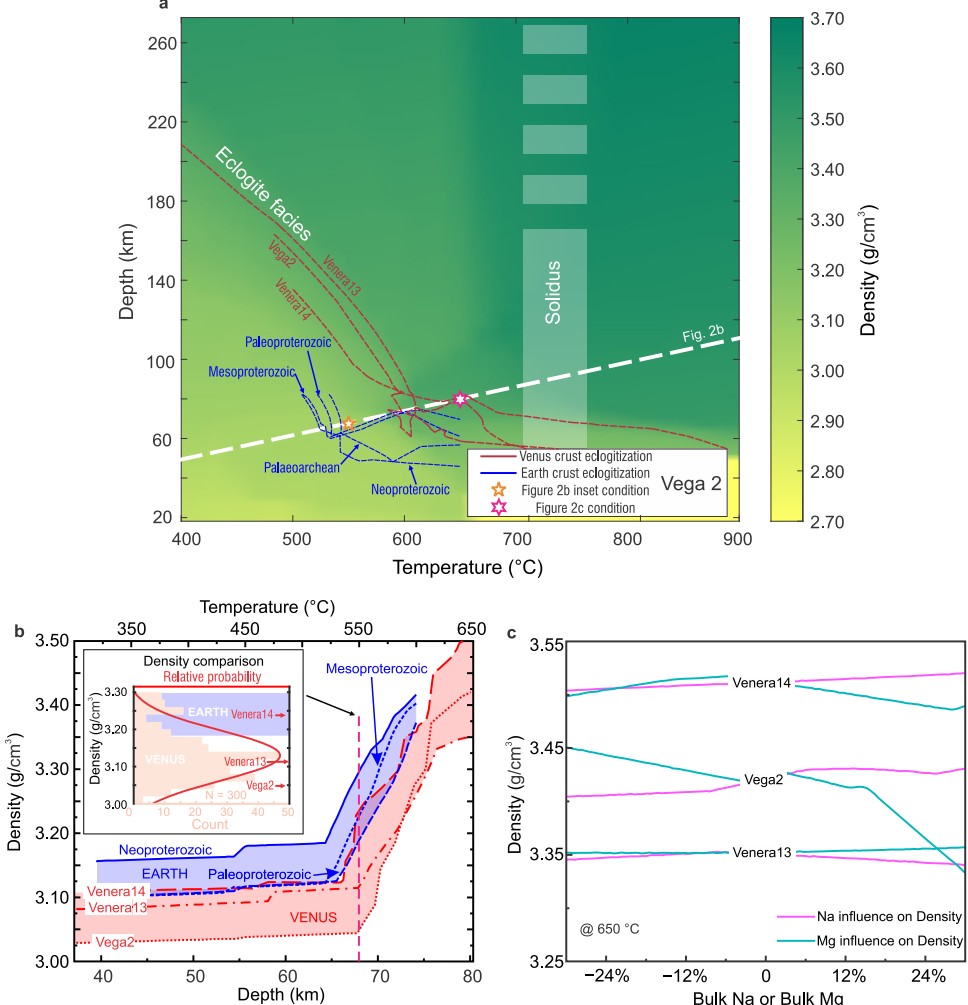

**Fig. 2 | Modeled densities of slab crusts. a** $P$–$T$ density map in the range 400–900 °C, 20–270 km, calculated for Vega 2 composition (Supplementary Table 2). The corresponding phase diagram is presented in Supplementary Fig. 1. The blue and red dash lines display the $P$–$T$ boundaries of eclogite facies in Earth's and Venus' crusts. The white area at 700–750 °C broadly denotes the solidus of metabasites[38–41]. The bright yellow colors represent low densities, and dark green colors represent high densities. **b** The Earth[43] (blue) and Venus (red) densities of slab crust comparison along the $P$–$T$ path along the white dash line in Fig. 2a. The inset diagram in Fig. 2b shows the statistical distribution of Venus' density of slab crust at 550 °C (see Method). **c** The densities of slab crust as functions of the variations in bulk Mg (cyan) and Na (magenta) concentrations at 650 °C. The phase diagrams are presented in Supplementary Figs. 4 and 5.

algorithm. For simplicity, we ignored the radiogenic heat, the change in mantle temperature (see Methods), and other advective heat fluxes. The asthenosphere mantle is assumed to be isothermal, and the temperature was set at 1550 °C[37], which is the potential temperature on Archean-to-Paleoproterozoic Earth. The geotherm in the lithosphere was simplified as varying linearly with depth (Fig. 1b inset). As the slab rotates, its boundaries reach thermal equilibrium with the pre-set geotherm (Dirichlet boundary conditions). The mafic crust starts to melt when its temperature reaches the solidus (700–750 °C[38–41]; Supplementary Fig. 1). We progressively removed the melted portion (>750 °C) from slab density calculation to represent the thermal erosion of slab[42].

We modeled Earth's and Venus' crustal assemblages and densities assuming chemical equilibrium (e.g., Fig. 2a; See Methods), which is dependent on the $P$–$T$ conditions as well as the crustal composition. The crustal compositions compiled by ref. 43 are used to represent early Earth. On Venus, three bulk-rock compositions were measured by USSR landing probes in the 1980s: Venera 13[44], Venera 14[44], and Vega 2[45]. These data were acquired by X-ray fluorescence (XRF) and reflected a broadly basaltic crust of Venus. Light elements (e.g., Mg) had large uncertainties. In particular, Na concentrations were not

effectively measured but estimated using K, Mg, and Fe oxide concentrations[44,45]. Isostatic compensation models are not able to identify major heterogeneity within Venus' crust by fitting topography and gravity data[28]. Without further constraints, we used these three data points to represent the crustal compositions of Venus in the phase-equilibria modeling, and then tested the sensitivity of our results to Na and Mg concentrations to account for their large uncertainties.

## Temperature pattern and thermal buoyancy

Hot edges grow, and a cold core is preserved as the slab is submerged into the hot mantle (Fig. 3a). At the end of the simulations, the temperatures of the bottom ~50 km and the frontal ~25 km of the slab are over 1000 °C, while the rest of crust remains solid (<750 °C; Fig. 3d). Thermally eroded slab forms a wedge-shaped area at the subduction front, around 250 km long and 20 km thick (Fig. 3). The crustal temperature increases significantly by more than 300 °C (Fig. 3a, d, Supplementary Video 1). The core of the slab mantle remains cool (<1000 °C) (Fig. 3a, d), so is denser than the ambient mantle by ~0.1 g/cm³ (Fig. 3). The cold core provides negative thermal buoyancy for subduction (Fig. 4). We adopt the same model

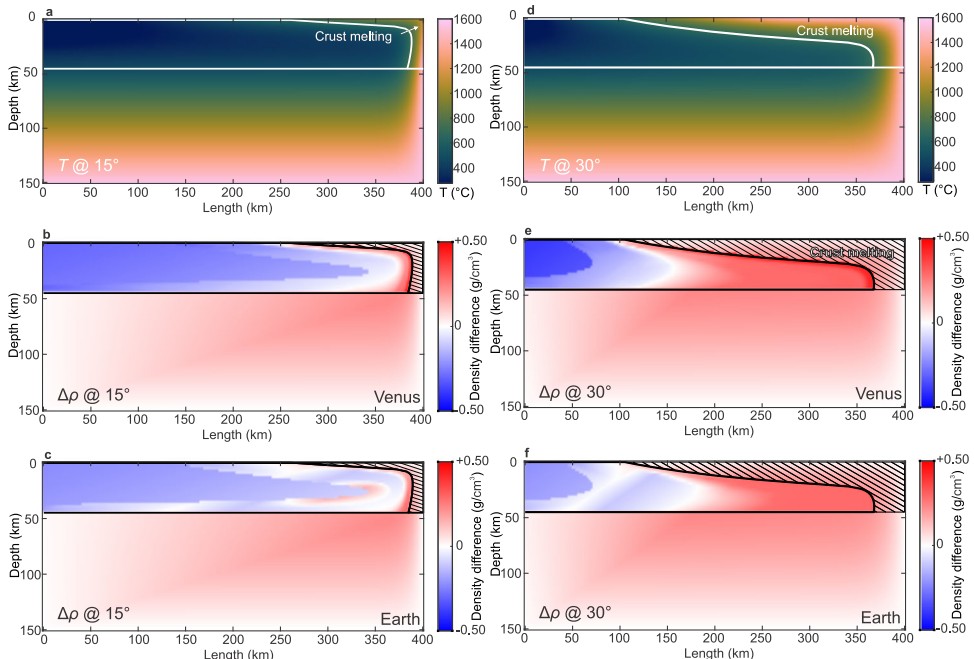

**Fig. 3 | Modeled 2-D maps of temperatures and densities of the slabs. a, d** Slab temperature pattern, when the submerged slabs rotate to 15° and 30°. Dark and cold colors represent low temperatures, and bright and warm colors represent high temperatures. **b, e** Paleoproterozoic Earth's slab density pattern, when the submerged slabs rotate to 15° and 30°. **c, f** Venus' (Vega2) slab density pattern, when the submerged slabs rotate to 15° and 30°. Blue represents a denser negative slab than the ambient mantle (negative buoyancy), and red represents a lighter slab. The modeling results are also provided as Movies in the Supplementary Information.

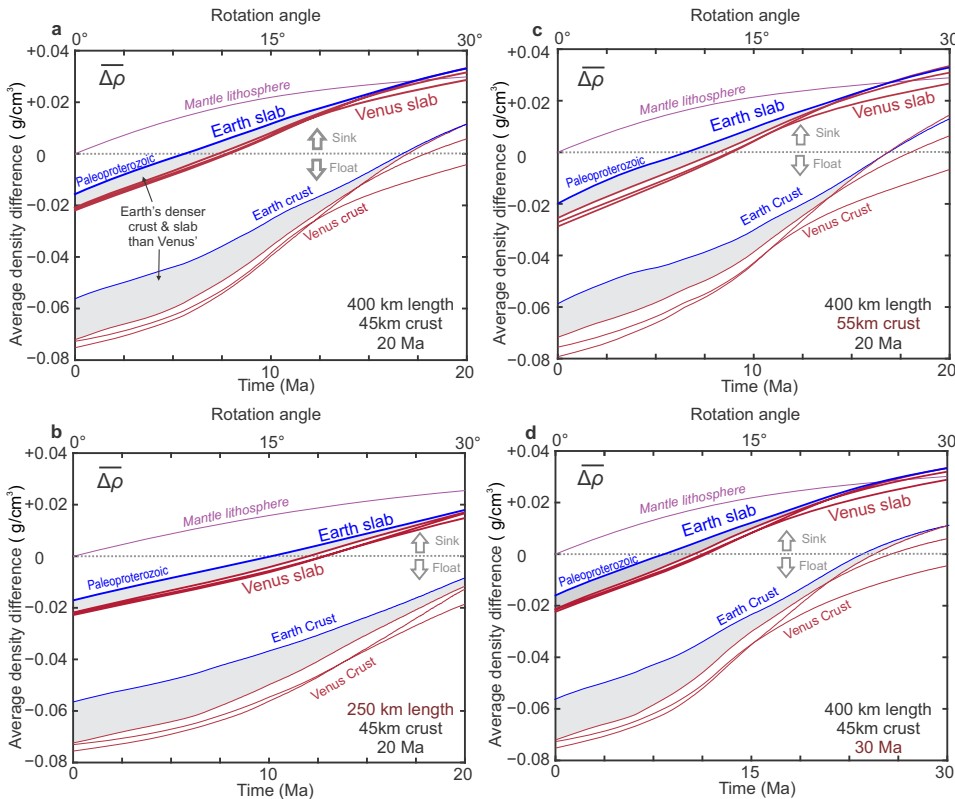

**Fig. 4 | Average density differences between ambient mantle and slab in various model settings. a** The model setting as Fig. 1b. 400 km long slab, 45 km crust, 105 km thick mantle lithosphere slab, subducted to 30° in 20 Ma. **b** 250 km long slab subducted to 30° in 20 Ma. **c** 55 km crust slab subducted to 30° in 20 Ma. **d** 400 km long slab rotated to 30° in 30 Ma. The Paleoproterozoic crust (blue) is the densest among Earth's (Fig. 2b) and is presented in this figure for comparison.

parameters for both planets, so the only minor difference in slab mantle density arises from the slightly different gravitational accelerations. The change of thermal buoyancy with pressure is not comparable to that with the metamorphosed crust.

## Crustal densification by eclogitization

The pressure–temperature (P–T) phase diagrams for the three Venus bulk compositions are topologically alike (Fig. 2a and Supplementary Figs. 1–3). The eclogite-facies assemblages consisting of dense minerals (clinopyroxene and garnet) become stable at >1.5 GPa (~70 km) and >530 °C on Venus (green-shaded in Fig. 2a and Supplementary Figs. 1–3). Venus' crust density sharply increases when eclogite forms. We use the density at 400 °C to represent that of low-grade rocks. At sub-eclogite conditions (<70 km), crustal densities are <3.2 g/cm$^3$ (Fig. 2a, b), while elogitization densifies the crust by over 0.3 g/cm$^3$ (Fig. 2a, b). A completely eclogitized crust is as dense as ~3.5 g/cm$^3$ (Fig. 2a, b).

Eclogitization occurs at a shallower depth and cooler temperature on Earth than those on Venus, so the Earth's crust becomes denser at the same depth as Venus' (Fig. 2a, b). The densities of Venus' crusts are consistently lighter than that of Earth's Proterozoic crusts by up to 0.1–0.2 g/cm$^3$ before and when eclogitization takes place (<650 °C; Fig. 2b) because Venus's crusts' higher Al contents result in more abundant low-density minerals during high-pressure metamorphism. During the submersion of a Paleoproterozoic Earth's slab into the mantle, its tip is first metamorphosed and becomes 0.1–0.2 g/cm$^3$ denser than the ambient mantle (Fig. 3b, c; Supplementary Video 2). This dense portion of the crust is ~20% smaller in Venus's slab (Fig. 3; Supplementary Videos 3–5). Additionally, the crust density in very low-temperature conditions (<400 °C) is ~0.1 g/cm$^3$ higher on Earth than on Venus (Fig. 2b). In summary, the chemical buoyancy of the metamorphosed crust is stronger in Venus's slab than early Earth, which hinders its sinking when subjected to external loading.

## Slab buoyancy in plume-induced subduction

The average density difference (see Method) between the ambient and slab mantles, which represents negative thermal buoyancy, are almost identical on Earth and Venus during subduction, because of the same assumptions regarding to the thermal structures and dynamical processes (Fig. 4a). Thus, the slab buoyancy difference is dominated by the metamorphosed crustal sections. Because of the greater depth of eclogitization in the slab, Venus' crust is significantly lighter than the Earth's (Fig. 2b), resulting in stronger buoyancy. The average difference between the ambient mantle and crust ($\rho_{crust} - \rho_{mantle}$), which represents positive chemical buoyancy, is consistently more pronounced by >0.02 g/cm$^3$ on Venus than on the early Paleoproterozoic Earth during the subduction process (Fig. 4a). Venus' lighter crust and accordingly lighter slab (Fig. 4a) lead to stronger resistance in the plume-induced subduction than that of Earth, making it less likely for the development of large-scale sustainable subduction.

## Additional modeling considerations: slab size and dynamics

The uncertainties in slab dimensions and the sinking rate have a minor influence on the slab buoyancy and our conclusions. In a shorter slab (250 km), both the Venus crust and slab are still lighter than those of the Paleoproterozoic Earth (Fig. 4b). With a smaller proportion of eclogite-facies metamorphism than in a longer slab (Fig. 4a), a shorter slab crust is always lighter than the ambient mantle on both Earth and Venus (Fig. 4b).

As the crust contributes to positive buoyancy and the mantle contributes to negative thermal buoyance, a slab with a lower crust/mantle ratio is more prone to sinking. As an example, if the slab crust is thicker (e.g., 55 km slab crust, 95 km slab mantle), the crust contributes more to the buoyancy, and the difference between the two planets' slab densities becomes slightly larger (Fig. 4c). A faster-

submerged slab preserves a larger cold and dense core in the mantle lithosphere. On the other hand, a smaller fraction of eclogite in the cold core leads to a lighter slab crust. The changes in thermal and chemical buoyancy largely cancel out. Reducing the sinking rate by 1/3 would not bring in observable differences in the densities of crust or lithosphere mantle in subduction and our conclusions remain unaffected (Fig. 4d).

## Additional modeling considerations: compositional effects

We now examine how the uncertainties in Venus' crustal compositions influence the equilibrium mineral assemblages and resultant crustal densities. We note that the density of metamorphosed Venera 14 approaches that of the Paleoproterozoic high-MgO basalt on Earth (Fig. 2b). Venera 14 is distinctively Mg-poor compared to the other Venus samples, and the effect of compositional variations on the crust density is discussed below in two ways.

In the first consideration, we assume that the three Venus samples with XRF measurements (Supplementary Table 2) sampled a uniform Venus' crust. The scatter in data was regarded as analytical uncertainties, so we propagate them into the crustal densities to determine the range of density variation (Methods). As an example, at 550 °C, where eclogitization starts, the distribution of the sampled densities peaks at 3.12 g/cm$^3$, with 1σ standard deviation of ±0.03 g/cm$^3$ (Fig. 2b inset); Venus' density is lower than Earth's is statistically significant.

We then evaluate the influences of Na and Mg contents on crustal densities by adding 30% variations in bulk rock Na and Mg contents in the phase-equilibria simulations (Fig. 2c and Supplementary Figs 4 and 5). Under the average crustal temperature of ~650 °C during subduction, the density variation due to the uncertainty associated with crustal Na content is minor (<0.05 g/cm$^3$, Fig. 2c). Mg and Fe$^{2+}$ substitute in mafic minerals, such as garnet and pyroxene. Mg-endmembers are less dense, resulting in lower rock density. The densities drop as the Mg contents increase, by up to 0.1 g/cm$^3$ (Fig. 2c). If the bulk Mg# of Venera 14 (0.62) were comparable to the others (Vega 2: 0.72, Venera 13: 0.69) by an increase in MgO by >30%, it would have a similarly low density. Venus has resurfaced repeatedly in the past[46,47], and therefore its surface rocks represent a relatively modern crust. In the early history of a planet, the melting of a hotter mantle would have produced crustal rocks with higher Mg contents[48] and a less dense slab under high-pressure metamorphism (Fig. 2c). In addition, the metamorphosed Neoproterozoic or modern MORB slabs are denser than those in the Paleoproterozoic (Fig. 2a, b) that are used to compare with Venus's slab. Thereby, our modeling results could be regarded as an upper limit for the crust densities on early Venus and a lower limit for the slab density difference between Earth and Venus.

## Implications for plate tectonics, environments, and future exploration

Using petrologic knowledge, we examined the slab densities and resulting buoyancies in hypothesized subduction on two planets. On Earth, when prototype subduction submerged a slab to mantle depth, the denser slab more likely triggered self-sustained subduction on a large scale[6,11,49,50] (Fig. 1c). The recycling of slab gave rise to continuous juvenile crust generation at convergent and divergent plate boundaries (~30 km$^3$/yr from mid-ocean ridges[51] and arcs[52]; Fig. 1c). On Earth, even in a sluggish tectonic regime argued to be operative in the Neoarchean (plate speed 20–30 mm/yr; ref. 25), a stable greenhouse and carbon cycle could still be maintained through the balance between silicate weathering and degassing[53]. On Venus, such self-sustained subduction is expected to be more difficult to develop due to the strong buoyancy of the slab crust. The resulting less active tectonic regime restrains continental lands[54,55] and likely limits fresh rock supply for weathering. Where plate tectonics is absent, fresh silicate rocks are mainly produced by plume-related volcanism at a rate

of ~0.37 km³/yr[47], about ~1/100 of those on modern Earth[51,52] and probably even less than the Archean Earth[56,57]. The difference in tectonic regimes and the resultant fresh rock supplies could fundamentally influence the long-term climate patterns on these two planets[54,58,59], although the links between fresh rock supplies and weathering feedback remain controversial[58,60–62]. During the subsequent evolution, the warming and anhydrous surface on Venus prevented the lithospheric lid from forming plate boundaries or subducting[6,16] and locked Venus in a stagnant/sluggish lid regime (Fig. 1d).

Our model illustrates the role of lithosphere petrology in regulating a planet's tectonic regime, and, potentially, its habitability. The interpretations of model results are a preliminary step toward revealing how lithospheric compositional differences could diverge the trajectories of the two planets' tectonic and environmental evolution. A similar mechanism might have prevented slab subduction on other terrestrial planets, like Mars[63]. In contrast to the numerous Mars missions and a large collection of Martian meteorites, there are only USSR missions in the 1980s that brought up three data points of Venus' crustal composition.

More constraints on Venus' crustal composition would not be available, nor a more comprehensive understanding of the Venus's tectonic and environmental evolution would be made until another landing probe is sent to Venus.

## Methods

### Conduction model
The 2-D slab temperature ($T$) is modeled by solving the heat equation:

$$\frac{\partial T}{\partial t} = \nabla \cdot \kappa \nabla T \tag{1}$$

where the thermal diffusivity ($\kappa$) is:

$$\kappa = \frac{k}{\rho C_p} \tag{2}$$

where $k$ is thermal conductivity (W/(m K)), $C_p$ is specific heat capacity (J/(kg K)), and $\rho$ is density (kg/m³).

The heat equation was solved using the Crank–Nicolson finite difference algorithm[64]. For numerical discretization, we used $200 \times 75$ spatial grids and 4000-time steps. Boundary nodes were set at the same temperature as the nearby ambient mantle in each time step. The parameters following the terrestrial data from ref. 65 were used for slab temperature calculation. For the thermal conductivity, ref. 66 fitted an expression of $k(T)$ to experimental data temperature dependence thermal conductivity (listed as Eq. 4 in ref. 65). An alternative expression from ref. 67, used to test the uncertainties, displays similar thermal conductivities in our model. Following the Eq. 10 in ref. 65, we used the $C_p(T)$ expression that was determined in ref. 68. The lithosphere and ambient mantle's densities are functions of temperature and pressure. The temperature's contributions to the mantle densities were calculated following Eq. 8 in ref. 65. The equation was based on ref. 69 coefficients of thermal expansion from 599 to 2100 K. The dependence of density and bulk modulus on the pressure was based on the dataset of ref. 70.

In a basaltic slab, the low U, Th, and K contents limit the radiogenic heat to less than $1\,\mu W\,m^{-3}$[71]. Over our model time of 20 Ma, the corresponding temperature increase at the core is <80 °C. This weak heating would result in slightly higher eclogite proportions in the slab and, accordingly, a denser slab, and smaller buoyancy. For simplification, we use fixed boundary values in the conduction model (Fig. 1b), given that the slab dips into the convecting mantle. However, the ambient mantle is refrigerated as the cold slab is submerged, by up to 100 °C within 20 km from the slab interface[72]. The cooler boundary

temperature would result in a less eclogitized and less molten crust and a denser ambient mantle. Thus, the ambient mantle cooling results in a larger buoyancy. In addition, we note that the mantle potential temperature varies between 1500 and 1650 °C from the Paleoproterozoic to Archean (1550 °C close to the lower bound in the model), which also depends on the model Urey's ratio[37,73]. Thus, the refrigeration of the ambient mantle and the uncertainties in potential temperature might partially cancel out.

### Phase equilibria
Through thermodynamic and mass balance equations, the minerals + fluid equilibria and their compositions were calculable for a fixed rock bulk composition ($X$) at a specific pressure ($P$)–temperature ($T$) conditions. The equilibrium phase diagrams, also known as "pseudosection", depict the phase relations and outline the areas of stable phase assemblages in $P$–$T$, $T$–$X$, and $P$–$X$ spaces. The equilibrium phase diagrams were calculated using THERMOCALC version 3.33 (https://hpxeosandthermocalc.org/the-thermocalc-software/), a thermodynamic dataset of ds55[74] and compatible activity models (Supplementary Table 3). We did not include silicate melts due to a lack of compatible activity models. For simplicity, we regarded 750 °C as an effective solidus temperature (see compilation in Supplementary Fig. 1) and removed supersolidus crust from the density calculation.

We used the NCKFMASHTO (Na$_2$O–CaO–K$_2$O–FeO–MgO–Al$_2$O$_3$–SiO$_2$–H$_2$O–TiO$_2$–O$_2$) model system. The crust compositions of Venus are represented by three available XRF data (Supplementary Tables 1 and 2) from USSR landing probes Venera 13[44], Venera 14[44], and Vega 2[45]. The Venera 13 composition is an alkaline basalt likely consisting of weathered olivine, leucite, and nepheline. Venera 14 and Vega 2 are both weathered N-MORB-like basaltic tholeiite. We assumed that a liquid ocean existed on early Venus, like Earth, so the basaltic crust should have been hydrated. Progressive dehydration reactions during metamorphism keep H$_2$O in saturation. The crustal Fe$^{3+}$/ (Fe$^{2+}$ + Fe$^{3+}$) ratios are mainly controlled by the mantle redox condition and partial melting process. Assuming similar settings of the early Earth and Venus, we used the same ratio as the early Earth (Fe$^{3+}$/(Fe$^{2+}$ + Fe$^{3+}$) = 0.1 in mole)[43]. The three Venus data all contain a fair amount of sulfur, which might reflect anhydrite (CaSO$_4$) or sulfate-bearing scapolite[75], the products of chemical weathering by an extreme atmospheric SO$_2$ content. If Venus had the liquid ocean in its early stage, the sulfur content would not have been as significant. Thus, we ignored the S contents in our model. We did not remove corresponding CaO from the bulk composition because sulfur weathering does not introduce additional CaO. The MnO contents are low, have large uncertainties, and barely affect the phase relations, so we omitted MnO in the model system. The other elements' mass percentages were converted to mole fractions for the calculation.

Equilibrium phase diagrams predict the mineral assemblages and their compositions in a $P$–$T$–$X$ space (see Supplementary Note 1). We calculated pressure–temperature ($P$–$T$) equilibrium phase relations within the $P$–$T$ ranges of 0.5–7.5 GPa, 300–900 °C (Supplementary Figs. 1–3). The stable mineral assemblages and compositions were used to estimate Venus' crustal densities in subduction. To examine the influences of the less accurately measured Mg and estimated Na contents, we calculated $P$–$X_{Mg}$ and $P$–$X_{Na}$ equilibrium phase diagrams at 650 °C, 0.5–2.5 GPa, against ±30% uncertainties of Na and Mg bulk concentrations (Supplementary Figs. 4 and 5).

### Density model
The slab's 2-D density patterns were mapped from the $P$–$T$ density diagrams, using the modeled temperature patterns and the depths. At each $P$–$T$ condition, phase equilibria predict the fractions and compositions of phases. The rock densities ($\rho$) were calculated on the basis of volume fractions ($v_i$) and densities of phases ($\rho_i$), using a similar

method as ref. 76:

$$\rho = \sum_{i=1}^{n} \rho_i \nu_i \qquad (3)$$

where the density of a mineral was calculated using:

$$\rho_i = \frac{\sum_{j=1}^{m} M_j x_j}{\sum_{j=1}^{m} V_j x_j} \qquad (4)$$

where $M_i$ is the molar weight of an end member, calculated from its formula; $x_j$ is the mole fractionation of one end member, provided by the phase equilibria modeling results; $V_j$ is the molar volume of an endmember at specific $P$–$T$ conditions, calculated from the equation of state in ref. 70.

## Venus crust density distribution at 550 °C

We propagated the uncertainties of concentrations, and calculated Venus' crust density and its standard deviation, to explore how likely the crust density of Venus is to overlap that of Earth. This exercise was time-consuming, so we only calculated at a representative condition (Fig. 2a) where the calculated crust densities of Venera 14 overlap the densities of Earth's Proterozoic crust (Fig. 2b inset). The means and standard deviations of Venus' crustal bulk compositions are shown in Supplementary Tables 1 and 2. At each step, we randomly sampled one element's abundance according to its Gaussian distribution at a time, modeled the phase relations using the new bulk composition, and calculated the density of the assemblage. We performed 300 such samplings, and then fit the series of calculated densities into a normal distribution for the mean and standard deviation of the density estimate.

## Average density difference

The slab buoyancy is directly proportional to the density difference between the ambient mantle and the slab. We integrated the density difference in our modeled 2-D slab area and divided it by the modeled area for the average density difference:

$$\overline{\Delta\rho} = \int \left( \rho_{\text{slab}} - \rho_{\text{mantle}} \right) \mathrm{d}S / \int \mathrm{d}S \qquad (5)$$

where the mantle density is also a function of pressure and temperature at the corresponding depth. The average density difference reflects the slab buoyancy in the ambient mantle, with $\overline{\Delta\rho}$ denoting the negative buoyancy for the slab to sink, and vice versa. The average density differences were calculated separately to consider the contributions from the crust and mantle lithosphere through time (Fig. 4). Supersolidus crust is excluded from the integral.

## Data availability

The authors declare that all data supporting the findings of this study are available within the article and its supplementary information files.

## Code availability

The MATLAB code used to generate the thermal and density models reported in this study are available from the corresponding author upon request.

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

## Acknowledgements

This work was supported by the NSERC Discovery Grant (RGPIN-2018-03925) to X.C and the National Natural Science Foundation of China (grant 41888101). J.H. wants to acknowledge the financial support from the Chinese Academy of Sciences Pioneer Hundred Talents Program and the CIFAR Azrieli Global Scholarship. M. Tian acknowledges a CSH Fellowship from Universität Bern.

## Author contributions

X.C. initiated the idea, J.C. performed the simulation, and J.C. and X.C. wrote the first paper. H.J., M. Tang, J.H., and M. Tian all contributed to data interpretation, paper writing, and revision.

## Competing interests

The authors declare no competing interests.
