## [Peer Review File · Nature Communications]

Reviewers' comments:

Reviewer #1 (Remarks to the Author):

This paper presents phase equilibrium modeling of an estimated Venusian crustal composition along representative potential subduction geotherms. The density of a downgoing plate is calculated to determine if and when a subducting slab would become negatively buoyant with respect to the mantle. The paper argues that due to differences in the composition of Venusian crust (derived from Venera & Vega measurements), Venusian slabs are more buoyant than slabs on Earth, and remain positively buoyant relative to the mantle to greater depths. The paper argues this difference precludes plate tectonics on Venus but would allow it to develop on Earth. The paper further discusses how a lack of plate tectonics may have affected climate evolution via weathering and the carbonate-silicate cycle.

I think the core result of this work, the phase equilibrium modeling, and how seemingly subtle differences in crust composition between Earth and Venus can lead to differences in crustal density, is an interesting result that should be published. However the paper in present form has some significant issues that need to be addressed first. The interpretations of this result in terms of the operation of plate tectonics and climate evolution/habitability have a number of weaknesses that significantly call into question the ultimate conclusions this paper draws. There is also a major technical issue with the slab density calculations, or at least a lack of information that makes it near impossible to evaluate how that calculation was done. This will take a moderate to major revision, but with all of these issues fixed I do think the paper will be suitable for publication in Nature Communications.

Major/big picture comments

1. The most important issue is how slab density is calculated. The paper shows phase equilibria calculations for a Venusian crustal protolith undergoing metamorphism, and uses this to calculate crustal density. But the density of a slab includes not just the crust, but the underlying mantle lithosphere as well. In order to calculate slab density we need to also know the crust thickness, mantle lithosphere thickness, and temperature profile through the mantle lithosphere. I couldn't find any information in the text (including methods section or supplement) about what was assumed for crust thickness and mantle lithosphere thickness and temperature. It is essential that this information be included in the methods section; one cannot evaluate the applicability of the slab density calculations without knowing what is assumed about the crust and mantle lithosphere thicknesses, and temperature profile through the slab.

If it turns out that the paper actually is not including mantle lithosphere density in the slab density calculation, then the work cannot be published until the mantle lithosphere is included in the calculation. Cold mantle lithosphere is an essential driving force for mantle convection and subduction and cannot be ignored.

2. The paper argues that the density difference in slabs between Earth and Venus would preclude subduction on Venus and allow it on Earth. Looking at Figure 2, for the cold geotherm, slabs on Earth become negatively buoyant at 65-70 km depth, while on Venus it is 70-75 km. For the hot geotherm neither reach negative buoyancy for the range plotted, but trying to extrapolate the difference is maybe 10-15 km between the two. The difference does not seem very large! So why does it prevent subduction on Venus, yet allow it on Earth? The paper never justifies this. This is not a minor thing; the central claim of the paper is largely unsupported because there is no demonstration that the modeled density difference is enough to prevent Venusian subduction. The dynamics need to be more fully explored here to justify the paper's core claim.

3. Part of the paper's motivation for looking for a new explanation for a lack of plate tectonics on Venus is correctly pointing out that it is possible Venus had a temperate climate and liquid water oceans throughout a significant portion of its early history. This is a great point about how to explain a stagnant lid, or at least lack of plate tectonics, on early Venus, IF we could confirm this tectonic history. However, we really have no idea what the tectonic state of early Venus was. This undercuts the motivation here, or at least I think the paper goes too far with its current motivation; we do not necessarily need to explain why Venus had stagnant lid tectonics when it might have liquid water,

because we really don't know that Venus was operating in a stagnant lid regime at the time. For example, see Byrne et al. 2021 "A globally fragmented and mobile lithosphere on Venus" for an argument for some sort of mobile lithosphere tectonics on Venus in the past (and it's very hard to determine just how far in the past from current constraints).

It is still scientifically interesting to look for ways to explain a lack of plate tectonics on Venus even with a temperate climate and water. But the paper needs to acknowledge the high degree of uncertainty over Venus' tectonic evolution.

4. The implications of tectonic state for the carbonate-silicate cycle and habitability put forth in this paper also has some issues. I have argued (Foley & Smye 2018 *Astrobiology*; Foley, 2019 *Ap. J.*) that stagnant-lid tectonics does not impede the carbonate-silicate cycle and habitability in a wide range of cases. The key issue is supply limited weathering, as this paper discusses (and I covered extensively in my 2018 paper). However, this paper needs to actually complete the calculation to make their case that Venus would not have enough fresh, weatherable rock.

The paper estimates a long-term average of ~ 1 Tmol/yr of CO₂ emissions on Venus, and $0.37 \text{ km}^3/\text{yr}$ of erupted lava. The paper then claims this would lead to weathering being supply limited. Supply limited weathering only occurs when the rate of CO₂ outgassing is higher than the upper bound on silicate weathering based on complete weathering of all available rock. An eruption rate of $0.37 \text{ km}^3/\text{yr}$ scales to $3.7e8 \text{ m}^3/\text{yr}$, and with a density of 2500 kg/m^3 , gives $9.25e11 \text{ kg/yr}$. I estimated a weathering demand (amount of CO₂ drawn down from complete weathering of 1 kg of rock) for basalt of 5.8 mol/kg in my 2019 paper (see also Kump 2018 *Philosophical Transactions* paper). This gives a supply limit to weathering of ~ 5 Tmol/yr. In other words, weathering would only become supply limited if the outgassing rate of CO₂ exceeded 5 Tmol/yr. So weathering on a stagnant lid Venus would not be supply limited based on the numbers in this paper, unless the paper wants to argue that less than 1/5 of the erupted basalt would actually be available for weathering (this may be possible, but is hard to estimate quantitatively to my knowledge).

I also disagree that recycling CO₂ back into the mantle is essential for the carbonate-silicate cycle and habitability, though this claim is widely made in the literature on this topic. In my 2018 paper we argue that CO₂ is not returned to the mantle after outgassing, and instead recycles through the crust. Metamorphic release of carbonated crust buried by lava flows becomes the key process driving continued CO₂ outgassing after CO₂ in the mantle becomes depleted.

My papers are of course not the final word on this topic, and I'm sure many of the conclusions can be argued with. But in light of this and others' recent work on the potential for habitability in a stagnant-lid regime, this paper needs to do more work to bolster it's argument that a stagnant-lid regime on early Venus would lead to the loss of the weathering thermostat and development of a hothouse climate.

Minor & detailed comments:

Line 34: At least as written, it sounds like the paper is saying that an accumulation of CO₂ in the atmosphere is what triggers a runaway greenhouse. While this is possible (Colin Goldblatt has a paper or two on this), the standard view is that the runaway is driven solely by exceeding a solar radiation limit (e.g. see Kasting 1988, and a really nice overview paper Nakajima et al. 1992). The runaway happens even with no greenhouse warming by CO₂, barring other complications such as cloud feedbacks or spatial variability in water distribution that have been explored more recently. A moist greenhouse state that leads to rapid photodissociation and H escape can occur just through high CO₂ levels, but is not the same thing as a runaway greenhouse (Kasting 1988 is the key paper that describes and differentiates between these two processes).

Lines 43-44: There is actually a growing line of thought (I don't know if it could be called a consensus yet, though) that Venus is not really in a stagnant-lid regime today. There is work on possible subduction at some Coronae (Davaille et al. 2017) and the lithospheric thickness as estimated from flexure observations is thinner than what would be expected from a stagnant lid (Borrelli et al., 2021 *JGR Planets*).

Lines 131-133: I don't understand the argument here about mantle lithosphere buoyancy not being important or not contributing to slab buoyancy. Any sort of convective downwelling develops as some "bump" or region where the thermal boundary layer is thicker than the surrounding boundary layer, and this drives downwelling due to the density difference of the boundary layer and the surrounding and underlying warm mantle interior. The relevant comparison is therefore slab density and underlying mantle density, not slab density compared to overriding plate density. This subduction zone like structure where one would be comparing downgoing and overriding slab density wouldn't even develop until subduction has started anyway, at which point there will be some length of slab down in the mantle pulling the plate at the surface.

Lines 167-168: The Driscoll & Bercovici paper cited here is a good paper, but doesn't look at supply limited weathering so I don't think it really supports the point the manuscript is trying to make here.

Brad Foley

Reviewer #2 (Remarks to the Author):

Review of Chen et al – Nature Communications

Title: "VENUS' LIGHT SLAB HINDERS ITS DEVELOPMENT OF PLANETARY-SCALE SUBDUCTION AND HABITABILITY"

This manuscript from Chen et al uses phase equilibrium models to investigate the slab buoyancy force of Venusian crust under subduction zone conditions. I read this manuscript with great interest because it tackles an important question – why, given their many similarities, does Earth have plate tectonics and Venus does not?

We know very little about Venusian crust and even less about Venusian mantle. On one hand, this makes a study like this straightforward with limited compositionally heterogeneity to contend with. On the other hand, a lot of assumptions need to be made, such as the bulk density of Venusian upper mantle, and perhaps more importantly the bulk composition of Venusian crust prior to the relatively late resurfacing.

The methods employed here are sound insofar as forward modelling can be done with such limited control on Venusian crust compositions. I particularly like the authors' approach to dealing with uncertainty in Na and Mg contents.

I do, however, have concerns with how these results are interpreted. The authors are viewing the relative density between Venusian crust and hypothetical mantle as the primary deterrent in slab buoyancy. This is too simplistic of an approach because 1) it only considers the crustal portion of the slab, not the lithospheric mantle, and 2) it ignores the (greater) effect of slab thermal expansion on slab buoyancy. There is a range of thermomechanical literature that covers this subject, I would point to Weller et al, 2019 (EPSL) as one recent example that integrates thermomechanical analysis with petrological modelling. Even if the big assumptions made in this study about bulk crust and mantle compositions prove to be correct, the small density difference (in comparison with terrestrial compositions) are unlikely to be the chief determinant of whether Venusian slabs will subduct.

Figure 2a (and especially 2b) illustrates how little of an effect crust composition might have on slab buoyancy when you consider that the difference to depth of negative (crust vs upper mantle) buoyancy is only ~10km. Ignoring the point I raise above about thermal effects on slab buoyancy, if you can manage to get basaltic crust down to ~65km on Earth to begin subduction, it isn't a stretch to get Venusian crust to ~75km, especially if this need only occur occasionally to initiate plate tectonics (which is an entirely different unresolved topic).

I do not intend to be too critical of this study, there is value in the calculations made. However, I do not feel the results presented here and the methods used are suitable to make the bold claims written

here.

Reply to Dr. Foley's comments:

This paper presents phase equilibrium modeling of an estimated Venusian crustal composition along representative potential subduction geotherms. The density of a downgoing plate is calculated to determine if and when a subducting slab would become negatively buoyant with respect to the mantle. The paper argues that due to differences in the composition of Venusian crust (derived from Venera & Vega measurements), Venusian slabs are more buoyant than slabs on Earth, and remain positively buoyant relative to the mantle to greater depths. The paper argues this difference precludes plate tectonics on Venus but would allow it to develop on Earth. The paper further discusses how a lack of plate tectonics may have affected climate evolution via weathering and the carbonate-silicate cycle. I think the core result of this work, the phase equilibrium modeling, and how seemingly subtle differences in crust composition between Earth and Venus can lead to differences in crustal density, is an interesting result that should be published.

We appreciate Dr. Foley's constructive review of our original manuscript, and support Dr. Foley's summary of the original manuscript.

However the paper in present form has some significant issues that need to be addressed first. The interpretations of this result in terms of the operation of plate tectonics and climate evolution/habitability have a number of weaknesses that significantly call into question the ultimate conclusions this paper draws.

We agree with Dr. Foley that our interpretations of the plate tectonics and climate evolution/habitability had a number of weaknesses. In the revised manuscript, we focus on the tectonic evolutions and shorten the discussion on habitability to one paragraph. We remove the quantitative estimates with regard to carbon fluxes, given the large uncertainties and the paper length.

There is also a major technical issue with the slab density calculations, or at least a lack of information that makes it near impossible to evaluate how that calculation was done. This will take a moderate to major revision, but with all of these issues fixed I do think the paper will be suitable for publication in Nature Communications.

Major/big picture comments

1. The most important issue is how slab density is calculated. The paper shows phase equilibria calculations for a Venusian crustal protolith undergoing metamorphism, and uses this to calculate crustal density. But the density of a slab includes not just the crust, but the underlying mantle lithosphere as well. In order to calculate slab density we need to also know the crust thickness, mantle lithosphere thickness, and temperature profile through the mantle lithosphere. I couldn't find any information in the text (including methods section or supplement) about what was assumed for crust thickness and mantle lithosphere thickness and temperature. It is essential that this information be included in the methods section; one cannot evaluate the applicability of the slab density calculations without knowing what is assumed about the crust and mantle lithosphere thicknesses, and temperature profile through the slab.

If it turns out that the paper actually is not including mantle lithosphere density in the slab density calculation, then the work cannot be published until the mantle lithosphere is included in the calculation. Cold mantle lithosphere is an essential driving force for mantle convection and subduction and cannot be ignored.

We agree with Dr. Foley that the mantle lithosphere plays a critical role in slab buoyancy. Following both reviewers' suggestions, we used a simplified 2-D model thermal-metamorphic model that includes both the slab crust and mantle. This model involves fewer assumptions than the original manuscript. The new model setting is illustrated in Figure 1b. Given likely similar thermal structure and mantle dynamics on the two planets in the pre-plate-tectonic era (Line 67-73), we used same model parameters for the slabs (Line 76-81). The model parameters were varied in their likely ranges to assess how sensitively the conclusions depend on the model setting (Figure 4 and the section from Line 158).

2. The paper argues that the density difference in slabs between Earth and Venus would preclude subduction on Venus and allow it on Earth. Looking at Figure 2, for the cold geotherm, slabs on Earth become negatively buoyant

at 65-70 km depth, while on Venus it is 70-75 km. For the hot geotherm neither reach negative buoyancy for the range plotted, but trying to extrapolate the difference is maybe 10-15 km between the two. The difference does not seem very large! So why does it prevent subduction on Venus, yet allow it on Earth? The paper never justifies this. This is not a minor thing; the central claim of the paper is largely unsupported because there is no demonstration that the modeled density difference is enough to prevent Venusian subduction. The dynamics need to be more fully explored here to justify the paper's core claim.

The distinct depths in the previous version were due to a simplified result, so its large uncertainties made this threshold unreliable. In the new manuscript, we completely changed our model setting, so that most of the concerns no longer exist.

Our focus is that the slab buoyancy, or resistance to sinking, is consistently larger on Venus than Earth in the developing stage of subduction (Fig. 4). The more buoyant slab makes the plume-induced subduction less likely to form self-sustained subduction on Venus than the Earth. The main reason for buoyant Venus slab is the smaller proportion of eclogite-facies crust.

3. Part of the paper's motivation for looking for a new explanation for a lack of plate tectonics on Venus is correctly pointing out that it is possible Venus had a temperate climate and liquid water oceans throughout a significant portion of its early history. This is a great point about how to explain a stagnant lid, or at least lack of plate tectonics, on early Venus, IF we could confirm this tectonic history. However, we really have no idea what the tectonic state of early Venus was. This undercuts the motivation here, or at least I think the paper goes too far with its current motivation; we do not necessarily need to explain why Venus had stagnant lid tectonics when it might have liquid water, because we really don't know that Venus was operating in a stagnant lid regime at the time. For example, see Byrne et al. 2021 "A globally fragmented and mobile lithosphere on Venus" for an argument for some sort of mobile lithosphere tectonics on Venus in the past (and it's very hard to determine just how far in the past from current constraints).

It is still scientifically interesting to look for ways to explain a lack of plate tectonics on Venus even with a temperate climate and water. But the paper needs to acknowledge the high degree of uncertainty over Venus' tectonic evolution.

We agree with Dr. Foley that the early stage of the Venus is largely unknown and almost impossible to examine. The motivation of this work to test how the Venus slab would behave and compare with on Earth in preferable conditions (water abundant, mild temperature, plume-induced subduction). In the revised Introduction section, we are more careful with wording and more explicit about the model setting. Given the first-order similarity in size and composition with few direct constraints, we assume that the mantle dynamics and thermal structure were similar on the two planets before prototype plate tectonics operated.

4. The implications of tectonic state for the carbonate-silicate cycle and habitability put forth in this paper also has some issues. I have argued (Foley & Smye 2018 Astrobiology; Foley, 2019 Ap. J.) that stagnant-lid tectonics does not impede the carbonate-silicate cycle and habitability in a wide range of cases. The key issue is supply limited weathering, as this paper discusses (and I covered extensively in my 2018 paper). However, this paper needs to actually complete the calculation to make their case that Venus would not have enough fresh, weatherable rock.

The paper estimates a long-term average of ~ 1 Tmol/yr of CO₂ emissions on Venus, and 0.37 km³/yr of erupted lava. The paper then claims this would lead to weathering being supply limited. Supply limited weathering only occurs when the rate of CO₂ outgassing is higher than the upper bound on silicate weathering based on complete weathering of all available rock. An eruption rate of 0.37 km³/yr scales to 3.7e8 m³/yr, and with a density of 2500 kg/m³, gives 9.25e11 kg/yr. I estimated a weathering demand (amount of CO₂ drawn down from complete weathering of 1 kg of rock) for basalt of 5.8 mol/kg in my 2019 paper (see also Kump 2018 Philosophical Transactions paper). This gives a supply limit to weathering of ~ 5 Tmol/yr. In other words, weathering would only become supply limited if the outgassing rate of CO₂ exceeded 5 Tmol/yr. So weathering on a stagnant lid Venus would not be supply limited based on the numbers in this paper, unless the paper wants to argue that less than 1/5

of the erupted basalt would actually be available for weathering (this may be possible, but is hard to estimate quantitatively to my knowledge).

We remove quantitative estimates regarding the carbon cycle, and only qualitatively imply the link between the lack of plate tectonics and supply-limited weathering (Line 208–232). We significantly shorten the discussion on climate to focus on the implications on tectonics in the revised manuscript.

I also disagree that recycling CO₂ back into the mantle is essential for the carbonate-silicate cycle and habitability, though this claim is widely made in the literature on this topic. In my 2018 paper we argue that CO₂ is not returned to the mantle after outgassing, and instead recycles through the crust. Metamorphic release of carbonated crust buried by lava flows becomes the key process driving continued CO₂ outgassing after CO₂ in the mantle becomes depleted. My papers are of course not the final word on this topic, and I'm sure many of the conclusions can be argued with. But in light of this and others' recent work on the potential for habitability in a stagnant-lid regime, this paper needs to do more work to bolster its argument that a stagnant-lid regime on early Venus would lead to the loss of the weathering thermostat and development of a hothouse climate.

The new discussion is less determinative of carbon balance. We remove the discussion on CO₂ recycling back to the mantle through subduction. If this buried carbon is not recycled through delamination, it is released through subsequent metamorphism as Dr. Foley proposed (Line 221–225).

Minor & detailed comments:

Line 34: At least as written, it sounds like the paper is saying that an accumulation of CO₂ in the atmosphere is what triggers a runaway greenhouse. While this is possible (Colin Goldblatt has a paper or two on this), the standard view is that the runaway is driven solely by exceeding a solar radiation limit (e.g. see Kasting 1988, and a really nice overview paper Nakajima et al. 1992). The runaway happens even with no greenhouse warming by CO₂, barring other complications such as cloud feedbacks or spatial variability in water distribution that have been explored more recently. A moist greenhouse state that leads to rapid photodissociation and H escape can occur just through high CO₂ levels, but is not the same thing as a runaway greenhouse (Kasting 1988 is the key paper that describes and differentiates between these two processes).

This sentence is revised to avoid the causal relationship.

We agree that term 'runaway greenhouse', particularly in astrophysical studies, has a simple and quantitative meaning: the solar irradiation upon a planet exceeds the outgoing longwave radiation (OLR). By this definition, the planet surface temperature will keep increasing towards silicate melting and even silicate vaporization. Venus' atmosphere is hot but is likely in a steady-state, so it's not 'runaway' *sensu stricto*.

We used the word 'runaway' here as a positive feedback: high atmosphere CO₂ → high atmosphere temperature → → weak or no negative feedback to drawdown atmosphere CO₂ → atmosphere CO₂ keeps increasing via degassing. If such a positive feedback continues, it can lead to a scenario where the planet loses all its liquid ocean, creating the so-called "steam atmosphere" (e.g. Höning, 2021).

We understand that this steam atmosphere, despite a result of the positive feedback (or so-called 'runaway' greenhouse here), can be stable because a H₂O-rich atmosphere enhances OLR that can still balance solar irradiation. The same, the 'solar irradiation > OLR' runaway greenhouse could happen even without the contribution by CO₂. Thus, we take caution when using 'runaway greenhouse' (line 30, 213) to avoid the misunderstanding.

Lines 43-44: There is actually a growing line of thought (I don't know if it could be called a consensus yet, though) that Venus is not really in a stagnant-lid regime today. There is work on possible subduction at some Coronae (Davaille et al. 2017) and the lithospheric thickness as estimated from flexure observations is thinner than what would be expected from a stagnant lid (Borrelli et al., 2021 JGR Planets).

It is possible that there are some subductions at some Coronae on Venus, which supports the motivation for our model. In the revised manuscript we use the phrase "stagnant/sluggish lid tectonics with limited plate mobility and

recycling” following the classification of Lenardic et al. (2016) (Line 36-37). It does not contradict the possible prototype-subduction sites on a ‘stagnant-lid’ Venus. We simply mean ‘no plate tectonics’.

Lines 131-133: I don’t understand the argument here about mantle lithosphere buoyancy not being important or not contributing to slab buoyancy. Any sort of convective downwelling develops as some “bump” or region where the thermal boundary layer is thicker than the surrounding boundary layer, and this drives downwelling due to the density difference of the boundary layer and the surrounding and underlying warm mantle interior. The relevant comparison is therefore slab density and underlying mantle density, not slab density compared to overriding plate density. This subduction zone like structure where one would be comparing downgoing and overriding slab density wouldn’t even develop until subduction has started anyway, at which point there will be some length of slab down in the mantle pulling the plate at the surface.

We agree with Dr. Foley that the mantle lithosphere’s contribution to the slab buoyancy is not a minor. We include it in our new 2-D modeling in our revised manuscript.

Lines 167-168: The Driscoll & Bercovici paper cited here is a good paper, but doesn’t look at supply limited weathering so I don’t think it really supports the point the manuscript is trying to make here.

The Driscoll & Bercovici (2013) paper presents a box model in which the degassing, subduction, and erosion rates are linked to plate velocity. It does not specifically focus on supply-limited weathering. We rephrased the sentence to avoid implying that (Line 219–221).

Reply to Reviewer #2's comments:

This manuscript from Chen et al uses phase equilibrium models to investigate the slab buoyancy force of Venusian crust under subduction zone conditions. I read this manuscript with great interest because it tackles an important question – why, given their many similarities, does Earth have plate tectonics and Venus does not?

We know very little about Venusian crust and even less about Venusian mantle. On one hand, this makes a study like this straightforward with limited compositionally heterogeneity to contend with. On the other hand, a lot of assumptions need to be made, such as the bulk density of Venusian upper mantle, and perhaps more importantly the bulk composition of Venusian crust prior to the relatively late resurfacing. The methods employed here are sound insofar as forward modelling can be done with such limited control on Venusian crust compositions. I particularly like the authors' approach to dealing with uncertainty in Na and Mg contents.

In addition to dealing with the uncertainties in Na and Mg individually, we add another statistical consideration in the revised manuscript. The uncertainties of all components are integrated and propagated to the uncertainty in slab density (Line 186–191; Figure 2b inset).

I do, however, have concerns with how these results are interpreted. The authors are viewing the relative density between Venusian crust and hypothetical mantle as the primary determinant in slab buoyancy. This is too simplistic of an approach because 1) it only considers the crustal portion of the slab, not the lithospheric mantle, and 2) it ignores the (greater) effect of slab thermal expansion on slab buoyancy. There is a range of thermomechanical literature that covers this subject, I would point to Weller et al, 2019 (EPSL) as one recent example that integrates thermomechanical analysis with petrological modelling. Even if the big assumptions made in this study about bulk crust and mantle compositions prove to be correct, the small density difference (in comparison with terrestrial compositions) are unlikely to be the chief determinant of whether Venusian slabs will subduct.

We agree with the reviewer's comment. In the revised manuscript, we use the new 2-D model to include both the crust and mantle lithosphere contribution to the slab buoyancy. The model is built in a similar but slightly simplified way as Weller et al. (2019). We also discussed the uncertainties in the model and the influence of these uncertainties to the results (line 159–177; Fig. 4).

Figure 2a (and especially 2b) illustrates how little of an effect crust composition might have on slab buoyancy when you consider that the difference to depth of negative (crust vs upper mantle) buoyancy is only ~10km. Ignoring the point I raise above about thermal effects on slab buoyancy, if you can manage to get basaltic crust down to ~65km on Earth to begin subduction, it isn't a stretch to get Venusian crust to ~75km, especially if this need only occur occasionally to initiate plate tectonics (which is an entirely different unresolved topic).

Dr. Foley shared this concern. We no longer discuss the depth corresponding to the negative/positive buoyancy transition, or use it as a threshold for self-sustained subduction. The argument of our 2-D model is that the Venus slab is continuously more buoyant than the Earth in the preliminary stage of subduction. The Venus slab, when submerged in response to magmatic loading, experiences greater resistance to sinking. Thus, self-sustained subduction is less likely to develop on Venus.

I do not intend to be too critical of this study, there is value in the calculations made. However, I do not feel the results presented here and the methods used are suitable to make the bold claims written here.

REVIEWER COMMENTS

Reviewer #1 (Remarks to the Author):

Review of VENUS' LIGHT SLAB HINDERS ITS DEVELOPMENT OF PLANETARY-SCALE SUBDUCTION AND HABITABILITY

This is a revision of a submission I previously reviewed. The paper argues that differences in the composition of Venus's basaltic crust compared to Earth's, as well as the slightly lower gravity on Venus, lead to a delayed (that is, deeper) transition to eclogite. These factors all combine to make Venus' crust less dense than comparable Earth-like oceanic crust. The paper argues that this density difference could be enough to prevent subduction on Venus, while allowing it to develop on the early Earth. This potential difference in tectonic style is then argued to contribute to the different climate evolution of Venus compared to Earth as well.

The revised manuscript is significantly improved from the earlier version by including the density of the mantle lithosphere. However, there are still technical issues that need to be addressed, and the main interpretations aren't fully supported by the modeling results presented. The results presented can really only support a more modest (though still interesting in its own right) claim than what the paper makes, as I'll outline below.

Ultimately I think the paper would be suitable for publication after some moderate revision. Much more extensive dynamical modeling would be needed to really support the claim about the crustal buoyancy difference stopping subduction on Venus. This modeling is probably beyond the reach of this paper, which is focused on the metamorphic petrology of Venusian crust. But, I think keeping the focus on the crustal petrology and toning the conclusions down to what can actually be supported by the modeling this paper employs will still leave an interesting paper that can be acceptable for publication.

Major comments:

1. Like I said in the summary above, the conclusions are still too strong for what the results presented in the manuscript can really support. I think the manuscript conclusively shows that Venus' crust is lower density than Earth's, including a greater depth needing to be reached before eclogite forms. This could have important implications for subduction initiation, but the claim that this density difference is a significant factor in why Earth has plate tectonics and Venus doesn't can really only be supported with fully dynamic subduction & mantle convection models. This is the only way to determine if the density difference is enough to prevent subduction. As I and the other reviewer noted with the original manuscript, the difference in crustal density is not huge, and neither is the depth difference where eclogite forms. So it is not clear why this relatively subtle differences would prevent subduction on Venus and not on the early Earth. Despite improvements made in the modeling in this revised version, this point is still open.

The paper needs to either include dynamic modeling that really shows subduction being prevented by the subtle Earth-Venus crustal density differences, or tone the conclusions down to note that this density difference exists, and it might influence subduction. But really future geodynamic studies would be needed to show whether it actually prevents subduction in the end or not.

2. Including the density of the lithospheric mantle is a significant improvement in the revised manuscript. But how this is calculated and incorporated into the results is confusing to me. The main issue is splitting the two contributions, from crust and lithospheric mantle, up in figure 4. I can see the utility in plotting them that way, but the paper should also calculate the integrated density of the whole slab (as in Oxburgh & Parmentier, 1977; Davies, 1992; or Korenaga, 2006). It is the whole slab density, and whether this is larger or smaller than mantle below, that determines whether the slab can sink or not. While it is interesting to see the relative contributions of crust and lithospheric mantle to the total

slab density, and how they compete with each other, ultimately it is the integrated density through the whole slab that matters for subduction. This really needs to be calculated and shown to support the arguments this paper is making.

It is also unclear from figure 4 if the densities shown are taking into account crust/mantle ratio in the slab, or just densities averaged over the two regions separately. As the manuscript text discusses, the thickness of crust compared to thickness of lithospheric mantle is the key thing determining whether slabs can be negatively buoyant overall. But this effect isn't shown in figure 4, at least not as I understand the figure.

3. Fixed temperature boundary conditions on the top and bottom of the slab are not very realistic. As the slab heats up it also cools the mantle on either side as the slab sinks. To keep the temperatures on the slab top and bottom fixed would require fast mantle flow flushing hot mantle along the top and bottom of the slab, but sinking slabs typically set the speed at which mantle flows (i.e. they drag the surrounding mantle with them). It is better to treat temperature far from the slab as fixed, and allow the mantle in contact with the slab to cool some as heat flows into the slab by conduction. I know this manuscript cites McKenzie papers where these boundary conditions are used, but more modern slab temperature profile work shows that the slab top and bottom boundaries are not fixed to the interior mantle temperature (e.g. van Keken's modeling papers).

4. I still disagree with the discussion on the carbonate-silicate cycle and climate evolution, though I appreciate the changes made from the original submission in response to my comments. The current manuscript is more measured here, but still argues for supply limited weathering and hot climates forming due to a lack of plate tectonics, which I don't agree with. Lack of continents is proposed as a problem, but continents can form through non-plate tectonic processes, as has been proposed for the early Earth. There is also some evidence for continent like features on Venus, maybe even with felsic crust, though this is still uncertain (Gilmore et al 2017). Likewise, the silicate weathering feedback is found to operate even with small areas of exposed land (Abbot et al. 2012; Foley, 2015), and seafloor weathering has now been argued to provide a strong climate feedback on its own (Krissansen-Totton et al., 2017 & 2018).

The manuscript also argues that recycling of carbonated crust back into the mantle is needed to keep CO₂ from building up in the atmosphere in a stagnant-lid regime, but this is not the case. My work on this finds that CO₂ is rarely recycled into the mantle after being deposited in the crust by weathering. It metamorphically decarbonates before it can be recycled. This just leads to an outgassing flux that weathering can then balance and keep climate regulated. At the same time the metamorphic degassing flux is becoming significant, volcanic degassing from the mantle is declining as carbon has been removed from the mantle and built up in the crust.

I see that the Hoening et al. 2021 paper has this metamorphic degassing flux pushing the planet into a runaway greenhouse; this is puzzling because it is not clear how Venus had liquid water at first in their model, and why increasing CO₂ caused the water to evaporate. Most 1-D climate models find Venus to always be above the runaway greenhouse limit (e.g. Hamano et al 2013), such that liquid water oceans would never form, outside of a cloud-albedo feedback that Way proposes (though see the recent Turbet et al. 2021 paper arguing against this). Further, most 1-D climate models also find that increasing atmospheric CO₂ does not lead to a runaway greenhouse, but could lead to a moist greenhouse where water can escape from the upper atmosphere (Nakajima et al. 1992).

Minor Comments:

Line 17: I don't think "subduction" should be plural; the singular form always sounds right to me

Line 29: "Venus is featured be" should be "Venus features"

Line 32: I think it's more straightforward to compare Venus early climate to modern Earth, where we have good constraints on our climate. Since the main issue is just about whether Venus was always extremely hot or had a temperate period at some point. But this is very small issue in the context of the paper.

The paragraph on early Earth tectonics, lines 36-47, writes about one particular view of early Earth evolution as if this is the definitive theory. It is a perfectly fine view to take but needs to be qualified to note that this is just one of many models for how plate tectonics developed on Earth.

Lines 76-77: It is not straightforward to determine lithosphere thickness based on mantle temperature, as this depends on factors like mantle viscosity (lithosphere thickness is determined by the vigor of convection). So the assumptions underlying how the ~150 km lithosphere thickness is determined should be spelled out. I'm less knowledgeable about how the crust thickness is constrained, but key assumptions in this estimate should be briefly spelled out as well.

Line 87: I don't understand what the "depth barrier to subduction" is, and what the physical control on this is

Line 128: "become" should be "becomes"

Line 128: Is the 70 km depth for Earth or Venus (since they have different gravities)?

Line 129: the phrase about dramatic density increase doesn't fit well in this sentence to my reading

Line 135: I don't understand what Earth's crust is being compared to here. Is this saying Earth's crust is always denser than Venus' at every depth? Is there a particular depth where this comparison is being made? Writing just needs to be cleaned up to make the point clear.

Line 138-139: What does "100 km long crust" mean? Slab lengths are all longer than this in the models presented. Is this just talking about crust at 100 km depth? Or the crust at 100 km from the trenchward edge of the slab?

Line 142: Buoyant crust acts against the slab sinking, but whether it actually prevents slab sinking depends on mantle lithosphere thickness and the dynamics of mantle convection

Line 153: The density difference is defined as $\rho_{\text{crust}} - \rho_{\text{mantle}}$ on the previous line, so crust should have a negative density difference. So I think density difference should be reported as smaller (more negative) on Venus, or larger in magnitude to not have to worry about the sign difference.

Line 166: What is the 0.6 number? Crust-mantle ratio? But how is this defined...by volume, mass, area, etc.

Line 167: I don't know what "reversely" means here. Not sure it's a real world, either

Lines 203-205: I'm confused about the more Mg-rich crust being less dense, when e.g. Korenaga 2006 shows crust getting more dense as mantle potential temperature increases (Figure 9b).

Line 208: "plate tectonic" should be "plate tectonics"

Line 209: what is "strong buoyancy" referring to? The next phrase mentions light crust, so is this about mantle lithosphere? But mantle lithosphere helps subduction.

Line 213: "contributed" should be "contributes"

Figure caption for Figure 2: Caption refers to "slab densities" for figure 2b. But is this really a slab density (averaging or integrating over the whole slab thickness) or just the crust density?

Line 417: "slab during from" I think "during" should be deleted

Line 424: "were" should be "was" (or "is" and just keep everything present tense)

-Brad Foley

Papers referenced

Oxburgh, E. R., & Parmentier, E. M. (1977). Compositional and density stratification in oceanic lithosphere-causes and consequences. *Journal of the Geological Society*, 133(4), 343-355.

Davies, G. F. (1992). On the emergence of plate tectonics. *Geology*, 20(11), 963-966.

Abbot, Dorian S., Nicolas B. Cowan, and Fred J. Ciesla. "Indication of insensitivity of planetary weathering behavior and habitable zone to surface land fraction." *The Astrophysical Journal* 756.2 (2012): 178.

Krissansen-Totton, Joshua, and David C. Catling. "Constraining climate sensitivity and continental versus seafloor weathering using an inverse geological carbon cycle model." *Nature communications* 8.1 (2017): 1-15.

Krissansen-Totton, Joshua, Giada N. Arney, and David C. Catling. "Constraining the climate and ocean pH of the early Earth with a geological carbon cycle model." *Proceedings of the National Academy of Sciences* 115.16 (2018): 4105-4110.

Gilmore, M., Treiman, A., Helbert, J., & Smrekar, S. (2017). Venus surface composition constrained by observation and experiment. *Space Science Reviews*, 212(3), 1511-1540.

Hamano, K., Abe, Y., & Genda, H. (2013). Emergence of two types of terrestrial planet on solidification of magma ocean. *Nature*, 497(7451), 607-610.

Turbet, M., Bolmont, E., Chaverot, G., Ehrenreich, D., Leconte, J., & Marcq, E. (2021). Day-night cloud asymmetry prevents early oceans on Venus but not on Earth. *Nature*, 598(7880), 276-280.

Nakajima, S., Hayashi, Y. Y., & Abe, Y. (1992). A study on the "runaway greenhouse effect" with a one-dimensional radiative-convective equilibrium model. *Journal of Atmospheric Sciences*, 49(23), 2256-2266.

Reviewer #2 (Remarks to the Author):

Dear authors,

Thank you for taking the time to consider my comments and for undertaking a thorough reevaluation. A study like this requires one to make significant assumptions. While I disagree with some of your assumptions (e.g., that the Venera and Vega probes sampled uniform crust), they are clearly documented so other researchers can make their own evaluation if they so desire.

As an aside, your study nicely illustrates just how much value the tectonics community will gain by collecting new geological data from Venus. As you note, the difference in crustal

density between Earth and Venus is persistent yet rather minor, which (to me) makes their divergent tectonic histories all the more intriguing.

**Best,
Dr. Brendan Dyck**

Reply to Dr. Foley's comments:

This is a revision of a submission I previously reviewed. The paper argues that differences in the composition of Venus's basaltic crust compared to Earth's, as well as the slightly lower gravity on Venus, lead to a delayed (that is, deeper) transition to eclogite. These factors all combine to make Venus' crust less dense than comparable Earth-like oceanic crust. The paper argues that this density difference could be enough to prevent subduction on Venus, while allowing it to develop on the early Earth. This potential difference in tectonic style is then argued to contribute to the different climate evolution of Venus compared to Earth as well.

The revised manuscript is significantly improved from the earlier version by including the density of the mantle lithosphere. However, there are still technical issues that need to be addressed, and the main interpretations aren't fully supported by the modeling results presented. The results presented can really only support a more modest (though still interesting in its own right) claim than what the paper makes, as I'll outline below.

We'd like to express our sincere appreciation for Dr. Foley's constructive comments on our revised manuscript, and his efforts in helping us improve this study.

We agree with Dr. Foley's comments and understand his concerns that center on 1) the overstatement that the slab densities dominantly determine plate tectonics and environmental responses; 2) the focus on the crust component of a slab; and 3) the simplification of the model setups and assumptions, along with other minor and language issues in our manuscript.

Ultimately I think the paper would be suitable for publication after some moderate revision. Much more extensive dynamical modeling would be needed to really support the claim about the crustal buoyancy difference stopping subduction on Venus. This modeling is probably beyond the reach of this paper, which is focused on the metamorphic petrology of Venusian crust. But, I think keeping the focus on the crustal petrology and toning the conclusions down to what can actually be supported by the modeling this paper employs will still leave an interesting paper that can be acceptable for publication.

We adopted Dr. Foley's suggestion and toned down the claim on the link between metamorphism and self-sustained subduction, and further removed the text on weathering feedbacks. This study aims to illustrate the potential role of petrologic process in planetary evolution, and a much more extensive dynamic model is indeed beyond the scope. We kept the paper's main focus on the petrology and slightly modified the general idea, structure, claims, and figures. At the end of the abstract and the discussion section, we added a brief sentence/paragraph to emphasize the petrologic perspectives and the lack of reliable data from Venus.

The detailed replies are listed below.

1. Like I said in the summary above, the conclusions are still too strong for what the results presented in the manuscript can really support. I think the manuscript conclusively shows that Venus' crust is lower

density than Earth's, including a greater depth needing to be reached before eclogite forms. This could have important implications for subduction initiation, but the claim that this density difference is a significant factor in why Earth has plate tectonics and Venus doesn't can really only be supported with fully dynamic subduction & mantle convection models. This is the only way to determine if the density difference is enough to prevent subduction. As I and the other reviewer noted with the original manuscript, the difference in crustal density is not huge, and neither is the depth difference where eclogite forms. So it is not clear why this relatively subtle differences would prevent subduction on Venus and not on the early Earth. Despite improvements made in the modeling in this revised version, this point is still open.

The paper needs to either include dynamic modeling that really shows subduction being prevented by the subtle Earth-Venus crustal density differences, or tone the conclusions down to note that this density difference exists, and it might influence subduction. But really future geodynamic studies would be needed to show whether it actually prevents subduction in the end or not.

Dr. Foley pointed out that our last manuscript stating Venus's light slab crust due to a smaller eclogitized portion, and the lighter crust dominating the tectonic regime on Venus could be overly strong. We agree that our latter statement cannot be fully supported by the simplified model and a more sophisticated dynamic model would be required to address the lack of plate tectonics on Venus. Such a model, if available in the future, would require insights from petrologic studies as important constraints that we show have considerable uncertainties. The revised manuscript still focuses on the petrological perspectives and are more careful with our statements. We toned down our claims to be more moderate than in the previous version: eclogitization could have important implications for the slab buoyancy in subduction, but we do not argue that the high-pressure metamorphism is the decisive factor.

In the last manuscript, we removed the relative depths of eclogitization, which might be a leftover impression of Dr. Foley. The revised manuscript argues that the light slab crust on Venus (and the light bulk slab in the revised manuscript, see our reply to Point 2 below) makes slab submersion more difficult and less likely on Venus in a plume-induced process similar to that on Earth.

2. Including the density of the lithospheric mantle is a significant improvement in the revised manuscript. But how this is calculated and incorporated into the results is confusing to me. The main issue is splitting the two contributions, from crust and lithospheric mantle, up in figure 4. I can see the utility in plotting them that way, but the paper should also calculate the integrated density of the whole slab (as in Oxburgh & Parmentier, 1977; Davies, 1992; or Korenaga, 2006). It is the whole slab density, and whether this is larger or smaller than mantle below, that determines whether the slab can sink or not. While it is interesting to see the relative contributions of crust and lithospheric mantle to the total slab density, and how they compete with each other, ultimately it is the integrated density through the whole slab that matters for subduction. This really needs to be calculated and shown to support the arguments this paper is making.

It is also unclear from figure 4 if the densities shown are taking into account crust/mantle ratio in the slab, or just densities averaged over the two regions separately. As the manuscript text discusses, the thickness of crust compared to thickness of lithospheric mantle is the key thing determining whether slabs can be negatively buoyant overall. But this effect isn't shown in figure 4, at least not as I understand the figure.

We agree with Dr. Foley's comment that the integrated density difference between the slab and ambient mantle should be provided explicitly in Figure 4, in addition to individual slab mantle and crust. In the revised manuscript, we added the integrated average density difference of the whole slab to Figure 4, and illustrated the effect of crust/mantle ratio in Figure 4c. The variation of crust/mantle ratio within a reasonable range (discussion in Line 78-88) does not affect our general conclusion.

The densities of the slab mantle (and accordingly the density difference in Figure 4) are similar on the two planets, with a minor difference due to gravitational accelerations (Line 129-131), because we assume the same physical process and parameters (Line 69-73). Thus, the dominant factor in our model is the density of metamorphosed slab crusts.

3. Fixed temperature boundary conditions on the top and bottom of the slab are not very realistic. As the slab heats up it also cools the mantle on either side as the slab sinks. To keep the temperatures on the slab top and bottom fixed would require fast mantle flow flushing hot mantle along the top and bottom of the slab, but sinking slabs typically set the speed at which mantle flows (i.e. they drag the surrounding mantle with them). It is better to treat temperature far from the slab as fixed, and allow the mantle in contact with the slab to cool some as heat flows into the slab by conduction. I know this manuscript cites McKenzie papers where these boundary conditions are used, but more modern slab temperature profile work shows that the slab top and bottom boundaries are not fixed to the interior mantle temperature (e.g. van Keken's modeling papers).

The subduction of a cold slab does cool the ambient mantle. According to van Keken et al. (2002)'s model, the ambient mantle is cooled down by up to 100 °C within 20 km from the slab interface. However, among various simplifications, Dirichlet boundary conditions are used for the thermal model for a few reasons. (1) The heterogeneous mantle temperature gives rise to more complicated dynamics and considerations for buoyancy, as the cooler ambient mantle is denser. The denser mantle would provide greater buoyancy, but the denser and cooler halo itself would sink. (2) The cooling of the ambient mantle by slab might offset the uncertainties in the potential temperature. The mantle potential temperature varies between 1500-1650 °C from the Paleoproterozoic to Archean (1550 °C close to the lower bound in the model), which also depends on the model Urey's ratio (Korenage, 2008; Herzberg, 2010).

The incorporation of mantle temperature change might blur the focus of this model. Its effect on slab buoyancy appears of secondary importance/significance, and the most uncertainties in this study are associated with first-order factors such as the protolith composition. In the revised manuscript, we added an explanation paragraph (Line 474-481) that explicitly states the simplification.

4. I still disagree with the discussion on the carbonate-silicate cycle and climate evolution, though I appreciate the changes made from the original submission in response to my comments. The current manuscript is more measured here, but still argues for supply limited weathering and hot climates forming due to a lack of plate tectonics, which I don't agree with. Lack of continents is proposed as a problem, but continents can form through non-plate tectonic processes, as has been proposed for the early Earth. There is also some evidence for continent like features on Venus, maybe even with felsic crust, though this is still uncertain (Gilmore et al 2017). Likewise, the silicate weathering feedback is found to operate even with small areas of exposed land (Abbot et al. 2012; Foley, 2015), and seafloor weathering has now been argued to provide a strong climate feedback on its own (Krissansen-Totton et al., 2017 & 2018).

The manuscript also argues that recycling of carbonated crust back into the mantle is needed to keep CO₂ from building up in the atmosphere in a stagnant-lid regime, but this is not the case. My work on this finds that CO₂ is rarely recycled into the mantle after being deposited in the crust by weathering. It metamorphically decarbonates before it can be recycled. This just leads to an outgassing flux that weathering can then balance and keep climate regulated. At the same time the metamorphic degassing flux is becoming significant, volcanic degassing from the mantle is declining as carbon has been removed from the mantle and built up in the crust.

I see that the Hoening et al. 2021 paper has this metamorphic degassing flux pushing the planet into a runaway greenhouse; this is puzzling because it is not clear how Venus had liquid water at first in their model, and why increasing CO₂ caused the water to evaporate. Most 1-D climate models find Venus to always be above the runaway greenhouse limit (e.g. Hamano et al 2013), such that liquid water oceans would never form, outside of a cloud-albedo feedback that Way proposes (though see the recent Turler et al. 2021 paper arguing against this). Further, most 1-D climate models also find that increasing atmospheric CO₂ does not lead to a runaway greenhouse, but could lead to a moist greenhouse where water can escape from the upper atmosphere (Nakajima et al. 1992).

We agree that our statement about carbon and environmental evolution on Venus is beyond the direct support of our modeling results, and that quantitative feedbacks between silicate weathering and climate evolution remain a debated topic due to large uncertainties in modeling efforts.

In the revised manuscript, we further reduced the discussion about the habitability on Earth and Venus to avoid unsubstantiated claims. We revised the corresponding paragraph (Line 213-229) and pointed out the controversies from previous studies. We also rewrote the conclusion paragraph to emphasize the petrologic perspective and project how future Venus missions could help our understanding of Venus's petrological and tectonic evolutions (Line 231-239).

Minor Comments:

Line 17: I don't think "subduction" should be plural; the singular form always sounds right to me

Fixed.

Line 29: “Venus is featured be” should be “Venus features”

Fixed.

Line 32: I think it’s more straightforward to compare Venus early climate to modern Earth, where we have good constraints on our climate. Since the main issue is just about whether Venus was always extremely hot or had a temperate period at some point. But this is very small issue in the context of the paper.

We agree that the early-stage conditions are largely uncertain. As we rewrote the discussion about carbon and environmental evolution, the comparison in this regard between Venus and early Earth has been removed.

Dr. Foley’s comment reminds us of an irrelevant point to clarify. Earth’s modern slab crust is much denser than that of the Paleoproterozoic Earth, resulting in a denser slab (Fig. 2b), which is noted in Line 207-211.

The paragraph on early Earth tectonics, lines 36-47, writes about one particular view of early Earth evolution as if this is the definitive theory. It is a perfectly fine view to take but needs to be qualified to note that this is just one of many models for how plate tectonics developed on Earth.

The sentence is modified to reflect that this is just one of many theories about Earth’s plate tectonic evolution.

Lines 76-77: It is not straightforward to determine lithosphere thickness based on mantle temperature, as this depends on factors like mantle viscosity (lithosphere thickness is determined by the vigor of convection). So the assumptions underlying how the ~150 km lithosphere thickness is determined should be spelled out. I’m less knowledgeable about how the crust thickness is constrained, but key assumptions in this estimate should be briefly spelled out as well.

The thicknesses of lithosphere and crust and their likely ranges are explained with more details in Line 78-83.

Line 87: I don’t understand what the “depth barrier to subduction” is, and what the physical control on this is

The depth of “self-sustained” subduction is inferred from the petrologic studies in high-pressure metamorphic terranes. The text is revised to reduce ambiguity (Line 94). The point to make is that if self-sustained subduction could be started, the model with 30° slab rotation would be enough.

Line 128: “become” should be “becomes”

Fixed.

Line 128: Is the 70 km depth for Earth or Venus (since they have different gravities)?

For Venus. Specified in Line 137.

Line 129: the phrase about dramatic density increase doesn't fit well in this sentence to my reading
Changed to 'sharply increase'.

Line 135: I don't understand what Earth's crust is being compared to here. Is this saying Earth's crust is always denser than Venus' at every depth? Is there a particular depth where this comparison is being made? Writing just needs to be cleaned up to make the point clear.

We modified the language. Earth's slab crust is consistently denser than Venus's by 0.1-0.2 g/cm³ before and shortly after eclogitization takes place (< 650 °C). The fully eclogitized slab crusts have similar densities and density differences from the ambient mantle.

Line 138-139: What does "100 km long crust" mean? Slab lengths are all longer than this in the models presented. Is this just talking about crust at 100 km depth? Or the crust at 100 km from the trenchward edge of the slab?

We meant the frontal portion of the crust. The language is revised to "..., its tip is first metamorphosed and becomes 0.1-0.2 g/cm³ denser than the ambient mantle" (Line 148)

Line 142: Buoyant crust acts against the slab sinking, but whether it actually prevents slab sinking depends on mantle lithosphere thickness and the dynamics of mantle convection

We agree that the crust/mantle lithosphere thickness ratio influence the buoyancy of slab and we discuss it in Figure 4c and associated text (Line 176-178).

As we assume that the physical properties of the mantle and slab are the same on Venus and Earth in this simplified model, the dynamics of mantle convention would be expected to be the same, so the only difference is from the chemical buoyancy. Our study aims to test whether the chemical buoyancy would be different on the two planets.

Line 153: The density difference is defined as rho_crust-rho_mantle on the previous line, so crust should have a negative density difference. So I think density difference should be reported as smaller (more negative) on Venus, or larger in magnitude to not have to worry about the sign difference.

Wording changed to 'more negative'.

Line 166: What is the 0.6 number? Crust-mantle ratio? But how is this defined...by volume, mass, area, etc.

The number was from Gülcher et al. (2020 NatGeo), defined by thickness (volume). The number is removed from the text, and the crust-mantle ratio is discussed qualitatively in Line 177.

Line 167: I don't know what "reversely" means here. Not sure it's a real word, either

Removed

Lines 203-205: I'm confused about the more Mg-rich crust being less dense, when e.g. Korenaga 2006 shows crust getting more dense as mantle potential temperature increases (Figure 9b).

Mg-endmembers are lighter than Fe-endmembers, and the lighter Mg-rich metamorphic rocks is our modeling result (Fig. 2c). This is different from Korenaga (2006) that considers the melt-restite relationship at different T_P , and the “metamorphic slab crust” in this context is not the same as “crust” in Korenaga (2006).

To avoid misunderstanding, we revised the sentences (Line 207-208).

Line 208: “plate tectonic” should be “plate tectonics”

Fixed.

Line 209: what is “strong buoyancy” referring to? The next phrase mentions light crust, so is this about mantle lithosphere? But mantle lithosphere helps subduction.

The bulk density of the slab is discussed in the revised paragraphs above and Figure 4. The “strong buoyancy” here refers to the crustal portion, and is specified in the revised text (Line 221).

Line 213: “contributed” should be “contributes”

Fixed.

Figure caption for Figure 2: Caption refers to “slab densities” for figure 2b. But is this really a slab density (averaging or integrating over the whole slab thickness) or just the crust density?

Yes, we meant slab crust there. Caption fixed.

Line 417: “slab during from” I think “during” should be deleted

Fixed.

Line 424: “were” should be “was” (or “is” and just keep everything present tense)

Fixed.

Reply to Dr. Dyck’s comments:

Dear authors,

Thank you for taking the time to consider my comments and for undertaking a thorough reevaluation. A study like this requires one to make significant assumptions. While I disagree with some of your assumptions (e.g., that the Venera and Vega probes sampled uniform crust), they are clearly documented so other researchers can make their own evaluation if they so desire.

We appreciate Dr. Dyck’s suggestion on the 2-D modeling that improved our manuscript. We agree that our models have large and unquantified uncertainties because of very limited data from Venus. These uncertainties and assumptions are explicitly stated in the text. We toned down our claims in the new manuscript.

As an aside, your study nicely illustrates just how much value the tectonics community will gain by collecting new geological data from Venus. As you note, the difference in crustal density between Earth and Venus is persistent yet rather minor, which (to me) makes their divergent tectonic histories all the more intriguing.

We rewrote the last paragraph according to Dr. Dyck's idea, in which we suggest that future Venus' missions, especially landing probes, would be critical to our understanding of divergent tectonic and environmental histories.

Best,
Dr. Brendan Dyck